# A weighted constraint satisfaction approach to human goal-directed decision making

**Yuxuan Li**●*, **James L. McClelland**●*

Department of Psychology, Stanford University, Stanford, California, United States of America

* liyuxuan@stanford.edu (YL); jlmcc@stanford.edu (JLM)

**Data Availability Statement:** All relevant data and code for statistical analysis, model fitting, and plotting are available in this GitHub repository: https://github.com/Effie-Li/weighted-goal-satisfaction-public.

## Abstract

When we plan for long-range goals, proximal information cannot be exploited in a blindly myopic way, as relevant future information must also be considered. But when a subgoal must be resolved first, irrelevant future information should not interfere with the processing of more proximal, subgoal-relevant information. We explore the idea that decision making in both situations relies on the flexible modulation of the degree to which different pieces of information under consideration are weighted, rather than explicitly decomposing a problem into smaller parts and solving each part independently. We asked participants to find the shortest goal-reaching paths in mazes and modeled their initial path choices as a noisy, weighted information integration process. In a base task where choosing the optimal initial path required weighting starting-point and goal-proximal factors equally, participants did take both constraints into account, with participants who made more accurate choices tending to exhibit more balanced weighting. The base task was then embedded as an initial subtask in a larger maze, where the same two factors constrained the optimal path to a subgoal, and the final goal position was irrelevant to the initial path choice. In this more complex task, participants' choices reflected predominant consideration of the subgoal-relevant constraints, but also some influence of the initially-irrelevant final goal. More accurate participants placed much less weight on the optimality-irrelevant goal and again tended to weight the two initially-relevant constraints more equally. These findings suggest that humans may rely on a graded, task-sensitive weighting of multiple constraints to generate approximately optimal decision outcomes in both hierarchical and non-hierarchical goal-directed tasks.

## Author summary

Different problems require the consideration of different information sources, including often useful long-range, future information that may impact our immediate decisions. However, when future information is irrelevant to a key subgoal, it can be desirable to focus on achieving the subgoal first. We suggest that humans rely on appropriately weighting relevant information over irrelevant information to generate decision outcomes in both types of situations. We conducted behavioral experiments and fitted models of decision processes to understand to what extent people considered various task factors in choosing the initial path in different mazes, both when a simple maze occurred alone or

**Funding:** The author(s) received no specific funding for this work.

**Competing interests:** The authors have declared that no competing interests exist.

was embedded as an initial part in a larger maze. Our results show that people approximate the optimal decision outcomes in both tasks by modulating the weighting of different factors during planning, and that people who made more accurate initial path choices modulated these weightings more successfully than those who made less accurate choices.

## Introduction

A hallmark of human intelligence is our ability to carry out goal-directed behavior: our plans and actions are guided by long-term goals. For behavior to be effective towards achieving a goal, future information related to the goal often shapes many critical steps in our decision making. For example, when packing for an upcoming trip, the weather and our planned activities at the destination must be taken into account, even though the act of packing happens at a separate earlier time well before we reach the destination. Studies have shown that initial-stage decision making already considers future choice points and incorporates whole-path or aggregated future information in a decision tree [1–3]. Backward reasoning stemming from known conditions about a goal is even sometimes the optimal strategy in figuring out how to achieve the goal [4].

Yet efficient planning is also marked by the ability to break a problem into smaller problems, focusing in succession on a sequence of subgoals on the path toward the ultimate goal [5, 6]. To come back to our traveling example, deciding whether to take the train or an Uber to the departure airport is one such smaller problem that can be independent from future aspects of the plan, once the flight and departure time have been settled. Studies have found that humans are adept at hierarchical planning, able to use learned knowledge to construct subtasks in novel problems and decompose tasks efficiently [7, 8]. Modeling work has provided precise accounts of how humans optimally group states in an environment and efficiently construct hierarchical task representations [8–11]. Furthermore, neural imaging studies have also supported the idea that the brain naturally organizes incoming stimuli based on their underlying hierarchical structure, delineating brain regions that signal differences among subgoal contexts as participants progress through them [12, 13].

On the surface, the situations discussed above may seem to call for different planning processes. When future information is relevant to decision making at an early point in a plan, selecting appropriate actions requires considering more proximal information together with future information. But when future information is irrelevant to an immediate subgoal, optimal processing may be enhanced if we first decompose the overall task and then evoke an independent process to solve the subgoal without influences from irrelevant factors outside of the subtask context. However, many types of human sequential behavior that seem naturally characterized by a composition of independent, more atomic processes have been shown to rely on a degree of parallel consideration of multiple factors [14]. For example, optimally typing the sequence of letters in a word can involve hand movements that prospectively prepare for future letters [15], increasing overall speed and fluency. People also produce speech errors reflecting intruding influences from words after the target word [16], suggesting that current and future spoken words are being planned at the same time. More recent work has similarly argued that parallel, context-sensitive processes, rather than explicitly hierarchical or modular computations, underlie routine sequential action [17] and value-based decision making [18]. We may therefore also expect that humans solving an embedded subtask might exhibit a graded and non-exclusive focus on the subtask, with some degree of consideration of information outside of the subtask context even in situations where this information is irrelevant.

Here, we consider how a weighted constraint satisfaction process that simultaneously exploits multiple constraints can provide an alternative to the hierarchical task decomposition account of human planning and decision making. While we do not rule out that decomposition may occur in some situations, we suggest that flexible decision making in many problems may rely instead on the task-sensitive modulation of the degree to which different pieces of information are weighted in selecting the next action. For example, when solving an embedded subgoal in a larger task, a greater weighting of the subgoal-relevant information relative to the subgoal-irrelevant information can effectively approximate the optimal decision outcomes that would be generated by an explicit task decomposition process followed by exclusive subtask execution.

Our experiments explore this weighted constraint satisfaction account by considering the decision processes people engage in when selecting the first step in two related maze tasks, where the same maze navigation problem either appears as an isolated task or as an initial subtask embedded in a larger maze (Fig 1A and 1B). We now consider the two tasks in more detail, analyzing the task factors that might influence people's initial path choices and discussing the expected behavioral outcomes produced by the hierarchical task decomposition approach and the weighted constraint satisfaction approach.

We consider first the base task, in which participants are rewarded for moving a token along the shortest path from a starting point to a designated goal (Fig 1A). The task can be considered to have three parts: First move out from behind the internal wall near the starting point; then traverse to the other side of the environment; then approach the goal by moving behind the internal wall near the goal. The optimal overall path, however, is determined both by a starting-point-proximal factor (the starting position relative to the internal wall near the starting point, henceforth called the *myopic* advantage) and a goal-proximal factor (the goal position relative to the internal wall near the goal, henceforth called the *future* advantage). Since the optimal paths differ in the first action that needs to be taken, an optimal initial path choice must therefore be jointly constrained by both factors. Thus, if planning inherently involves the simultaneous consideration of multiple constraints, this task is perfectly suited to exploit it, and we would expect both the myopic and the future advantage to affect the choice of the first step and the time it takes to make it.

Fig 1B shows the same maze in the base task, now embedded as an initial part of a larger maze such that what was previously the goal location now corresponds to a subgoal location that must be reached prior to navigating to the final goal. In this more complex task, finding the optimal initial path still depends on the same two constraints, now associated with the starting point and the subgoal location. If people adopt a hierarchical task decomposition approach in this case, we may expect a longer processing time before choosing the initial action to reflect the time required to infer the subgoal and decompose the task into parts, prior to directing effort at solving the subgoal (Fig 1C, left panel). Moreover, the subsequent process of choosing the optimal initial direction within the subtask should be identical to that in the base task and unaffected by the position of the final goal (Fig 1C, right panel).

Alternatively, a weighted constraint satisfaction approach could be recruited to support decision making in the more complex task. As alluded to above, an approximation of the pattern of responding that would be produced under the hierarchical approach can be achieved by a greater weighting of the two subgoal-relevant constraints compared to the initially-irrelevant final goal in deciding the initial action (Fig 1D). For path selection to still be close to optimal, the weighting of the various factors must allow the two subgoal-relevant constraints to exert a near equal and jointly predominant influence in decision making. But a small residual weighting of the irrelevant final goal might be reflected in a subtle influence of this factor on the time course and outcome of selecting the initial path. To explore this possibility, we can

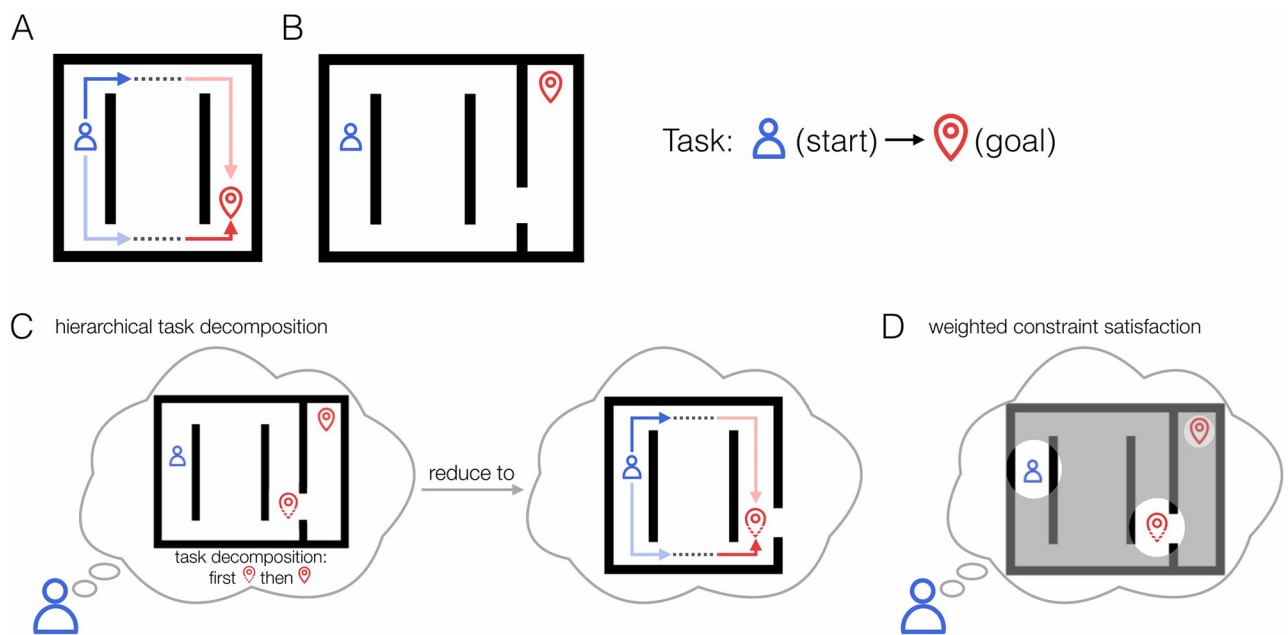

**Fig 1. Hierarchical and parallel planning processes. A**. A simple maze where selecting the initial direction of a best goal-reaching path benefits from the joint consideration of starting-point proximal (myopic) and goal-proximal (future) information. **B**. The task in A. embedded as an initial part of a larger maze. **C**. A hierarchical approach to selecting the initial path in B., by first reducing the task to the initial subgoal task then focusing on finding the best subgoal-reaching path. **D**. A parallel approach to selecting the initial path in B., where simultaneous but appropriately-weighted consideration of constraints both inside and outside of the subtask context generates approximately optimal choice behavior.

therefore ask these questions: 1) To what extent do participants balance their consideration of multiple *relevant* constraints, and 2) when there is an *irrelevant* constraint, to what extent do they effectively ignore it? More generally, how does the presence of a particular constraining factor (whether relevant or irrelevant) influence the weighting of other constraining factors?

Across two experiments, we studied how human participants approached solving these maze problems (Fig 2). We assessed both the paths chosen by participants, as indicated by the direction of their first step, and the time they took to plan this step, as indexed by their reaction time. We used the drift-diffusion model (DDM) [19] to characterize how the constraining factors discussed above were used in deciding the first action on each trial. The DDM and subsequent models built on related ideas have been used to understand a wide range of human perceptual and cognitive processes [20–22], and the DDM has recently been used to account for behavior in multi-step decision making [1, 3]. Here, we leverage the DDM to characterize whether the consideration of the various factors during planning was more consistent with a hierarchical task decomposition process or a weighted constraint satisfaction process. If planning occurs according to the hierarchical task decomposition approach as described above, the process of selecting the initial action in the more complex subgoal task should resemble that in the base task, though it may begin after a longer initial delay. In contrast, in the weighted constraint satisfaction approach, we might expect simultaneous influence of the subgoal-relevant constraints and the final goal during initial path selection, albeit with relatively less weight assigned to the final goal in decision making. We test these hypothesized decision processes by jointly fitting participants' path choices and response times.

In Experiment 1 (Fig 2, top panel), we first confirmed that initial path choices in the base task indeed reflected a joint influence from the relevant myopic and future constraints. Interestingly, we found that participants' path choices tended to reflect the myopic advantage more

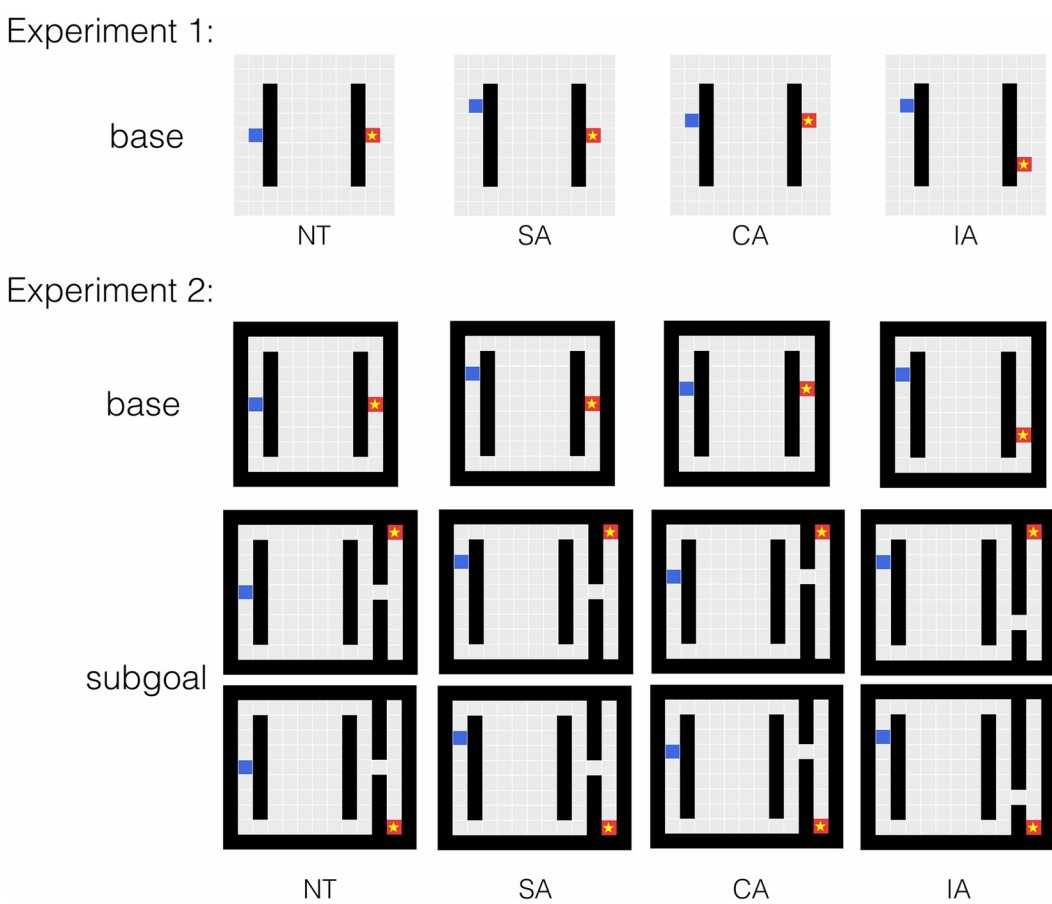

**Fig 2. Experimental design.** Blue block: starting point. Starred red block: goal location. Trial advantage types are based on the correspondence between the starting-point proximal constraint (myopic advantage) and the goal-/subgoal-proximal constraint (future advantage). NT = neutral advantage, SA = single advantage, CA = congruent advantage, IA = incongruent advantage. For the subgoal trials in Experiment 2, both of the possible final goal locations are shown.

strongly than the future advantage. In addition, when the myopic and future advantages were pitted against each other, each exerted a weaker influence on path choices than it did when only one of the factors was relevant to determining the optimal path. We then used the data from a subset of trials in this experiment to constrain the selection of a best-fitting DDM process for the base task. Then in Experiment 2 (Fig 2, middle and bottom panels), we compared the decision process in the more complex task with that in the base task. We found that initial path choices were slower in the complex task compared to the base task, and that these choices were both reduced in accuracy and influenced by the final goal position. In line with these observations, DDM variants consistent with the weighted constraint satisfaction account fit the pattern of data better than DDM variants consistent with the hierarchical task decomposition account. In addition, we found that more accurate participants (those who were more likely to choose the optimal path) assigned less weight to the initially-irrelevant final goal position than less accurate participants, and tended to exhibit more balanced weighting of initially-relevant constraints. In the *General Discussion*, we return to the idea that a parallel, weighted constraint satisfaction approach might be the biological brain's way of planning decisions, and that our brains approximate an optimal focus on relevant information in different task

contexts through the dynamic modulation of constraint weighting as thought and behavior unfold.

## Results

### Experiment 1

The two main goals of Experiment 1 were 1) to examine the influence of the starting-point proximal, *myopic* advantage and the goal-proximal *future* advantage in initial path choices in the base task and 2) to fit variants within the family of drift-diffusion models to a subset of the trials in the experiment to assess what constraint weighting scheme best captures the joint choice and response time patterns. We varied the myopic advantage and the future advantage, which together determine the shortest goal-reaching path on each trial, and analyzed participants' first action *choice* and their time taken to take the first step (*response time*).

**Methods**

*Ethics statement*. This study was approved by the Stanford University Institutional Review Board under protocol No.7029. In the online experiment, participants were first shown a consent page and were instructed to continue to the study if they agree to participate or exit at any time if they decline to participate in any or all parts of the study.

*Participants*. For Experiment 1, 100 US-based participants were recruited on Amazon Mechanical Turk. To ensure data quality, each participant must have had over 92% HIT approval rate and must have completed more than 1000 approved HITs to be eligible for the study.

*Task design*. Participants completed a single session consisting of three practice trials, 184 experimental trials, and a short survey. Each trial consisted of a shortest path search task on an 11×11 grid canvas with two internal walls (Fig 2, top panel). Participants were instructed to move the blue block to the starred goal location using the minimum number of up, down, left, or right steps. A step into the walls would increase the step count with no actual movement. Participants received a base completion compensation of $1.00 and a performance-based bonus on each trial ($0.03 if a trial was solved with the minimum number of steps, $0.01 if the solution was only up to two steps more than the shortest solution, and no bonus otherwise). The study took around 35 minutes and participants received an average of $6.05 for completing the study.

The experimental trials included 92 base trials and 92 filler trials, presented in a different randomized order for each participant. Below we report results from the base trials, where the locations of the starting block and the goal block were designed to vary the relative advantages of the candidate goal-reaching paths near the starting location and near the goal, but the locations of the two length-7 walls were fixed (see Table 1). Each unique base trial layout was mirrored vertically, except for the neutral trial. Both the original and the mirrored trials appeared in all four orientations, including the left-to-right orientation shown, as well as top-to-bottom, right-to-left, and bottom-to-top orientations.

The myopic advantage near the starting point refers to the side of the wall that the blue block can be moved toward to get out from behind the wall with fewer steps. Similarly, the future advantage near the goal refers to the side of the wall that the blue block can approach the goal from with fewer steps. Quantitatively, the myopic and the future advantages can be computed as the position offset of the starting block and the goal block relative to the center of the nearby wall. For example, in the trial shown in the top right panel of Fig 2, the myopic advantage is 2 (towards the upper path), and the future advantage is −2 (towards the lower path). The pairing of the two advantages establish four advantage types: neutral advantage (NT), single advantage (SA), congruent advantage (CA), and incongruent advantage (IA). We subset the SA and IA trials by whether the myopic or the future advantage was the larger

**Table 1. Myopic and future path advantages.** All trial layouts were mirrored except for the NT trial. NT = neutral advantage, SA = single advantage, CA = congruent advantage, IA = incongruent advantage. The -m and -f suffixes indicate whether the myopic or future advantage was larger.

| Adv. Type | Myopic Adv. | Future Adv. |
|:---:|:---:|:---:|
| NT | 0 | 0 |
| SA-m | 2 | 0 |
|  | 1 | 0 |
| SA-f | 0 | 2 |
|  | 0 | 1 |
| CA | 1 | 1 |
| IA | 1 | -1 |
|  | 2 | -2 |
| IA-m | 2 | -1 |
|  | 3 | -1 |
| IA-f | 1 | -2 |
|  | 1 | -3 |

advantage, denoted by the "-m" or "-f" tail. Note that in the NT trial layout and two of the IA trial layouts, the two main path candidates (i.e., equivalent to the upper and the lower paths in the left-to-right orientation) are equally optimal.

On each filler trial, we randomly sampled trial orientation (among all four orientations) and the length of each internal wall (3, 5, or 7). The two internal walls were also randomly shifted up or down, but were never allowed to block the upper or lower paths. Wall locations on the filler trials with two length-7 walls were also never centered (i.e., never identical to the wall locations on the base trials). The location of the starting block was randomly sampled from the locations to the left of the starting-proximal wall, and the location of the goal block was randomly sampled from the locations to the right of the goal-proximal wall.

*Exclusions*. We excluded data from five participants who were not able to complete the experiment due to technical reasons. Across all remaining 8740 observations (95 participants × 92 trials), we excluded trials where participants took longer than one minute to execute the first move (six trials) or took five or more steps compared to the longer of the two main path candidates (ten trials; a main path candidate is equivalent to the upper or lower path in the left-to-right orientation, without excessive steps). We also excluded trials with ill-identified initial path direction, including trials where the first move was equivalent to going left in the left-to-right orientation, or where the initial steps contradicted the overall path, e.g., an initial down action followed by a later upper path (31 trials).

*Drift-diffusion modeling*. We modeled the path choices and planning time in the IA trials as a drift-diffusion process, as these trials afford the opportunity to examine the influences of the two constraints when they each deviate from neutral and contribute to the decision outcome in opposite directions. The model treats the decision making process as involving a single aggregate decision variable in which a positive value favors a path choice that satisfies the myopic advantage and a negative value favors a path choice that satisfies the future advantage. On each trial, this variable evolves randomly over time with a mean direction $d$ given by:

$$d = md * mAdv - fd * fAdv \qquad (1)$$

where $md$ is the drift weight associated with the myopic advantage $mAdv$ and $fd$ is the drift weight associated with the future advantage $fAdv$. When the value of the variable reaches an

upper or lower bound at values $a$ or $-a$, the process terminates and a response satisfying the myopic advantage is chosen if the upper bound is reached, or one satisfying the future advantage if the lower bound is reached.

In addition to the parameters $md$, $fd$, and $a$, the model also includes an initial non-decision time $t_0$ and a possible starting point bias $z$. We additionally modeled inter-trial variability for the starting point ($sz$) and the drift rate ($sd$). As a baseline, we considered a model in which the weights for both advantages are equal ($md = fd$) and there was no starting bias ($z = 0$). We tested whether the data were better accounted for with equal or different advantage weights, with or without a starting bias, and with or without the two sources of inter-trial variability.

We pooled data from all participants to fit the candidate drift-diffusion models. We first z-scored the raw first move response time (in seconds) within each participant, then shifted the z-score distributions so that the minimum is 0.5 for each participant to ensure positive response time and enough non-decision time buffer for modeling purposes. All model variants were fitted in a Monte-Carlo cross-validation procedure written in R, using the density function implemented in the rtdists package [23] and the nlminb function in the stats package [24]. In each of 200 cross-validation folds, we held out data from 35 (out of 95) randomly sampled participants as the test data. Ten runs from random initial parameter values were optimized to minimize the summed negative log-likelihood (sNLL) of the training data; the number of runs per fold could be extended to avoid local minima (see S1 Text). The winning fit for each fold was selected based on the best training sNLL. Candidate models were fitted over the same 200 cross-validation folds to control for fold-level variability. For model comparison, we fitted a linear mixed-effects model to the sNLL on the held-out test data across all candidate models with fold-level random intercepts. We then selected as the "winning" model the one that had a reliably lower sNLL than other models or that had fewer free parameters than another model with a statistically indistinguishable sNLL. The winning model then served as the starting point for modeling individual differences and as the base model for Experiment 2.

*Analysis of individual differences.* To investigate how the weighting of constraints differed among participants, we split all participants into two groups based on the group median of initial path choice accuracy, where individual accuracy scores were computed on all trials with a unique optimal initial direction. We then fitted the best group model separately to data from participants with overall higher levels of accuracy (N = 51) and lower levels of accuracy (N = 44), using a cross-validation procedure identical to the group analyses, but re-sampling the test data within each group. We used a 36/15 train/test split for the higher accuracy group and 31/13 train/test split for the lower accuracy group. We then tested whether specific model parameter differences between the two groups could have arisen by chance from random assignment of participants into two groups. We conducted an additional round of fitting in which we compared the difference in fitted parameter values between the high and low accuracy groups to the distribution of differences observed between groups formed by 1000 random splits of the participants into two groups matched in size to the high and low accuracy groups. In this additional round of fitting, data from all participants in each group were used to train the model, so that the estimated differences reflected the data from all of the participants assigned to each group.

**Results and discussion**

Before turning to drift-diffusion modeling, we first present a descriptive analysis of the overall choice and reaction time data to confirm that participants' choices are influenced both by myopic and future constraints and to characterize the experimental factors that influenced the degree of choice optimality and the time required to make the choice.

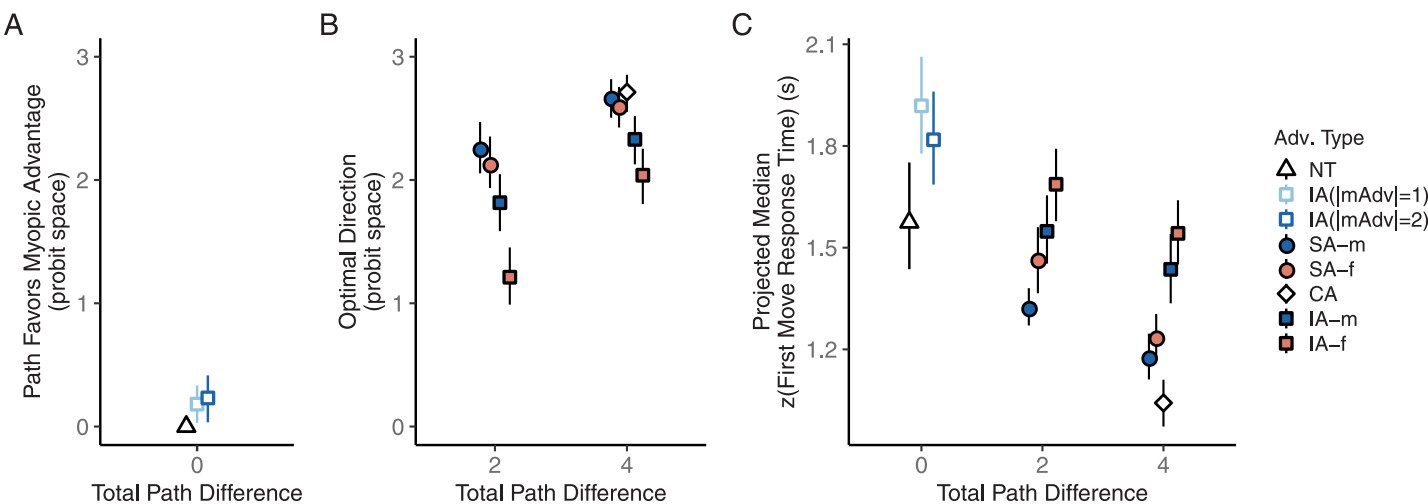

**Fig 3. Initial path choices and the associated response times reflected joint consideration of myopic and future constraints. A.** Path choices in trials with equally optimal initial directions. The value for NT trials was defined to be 0 since there was neither a myopic nor a future advantage on these trials. **B.** Path choices in trials where one of the initial directions was optimal. The proportion of optimal trials for each individual was converted into a probit score. If the individual probit score was larger than 3 or smaller than -3, it was capped at 3 or -3 before averaging. **C.** Response times associated with the first step. The individual medians of zscored response times were projected back to the raw time scale in seconds using group average mean response times and group average standard deviations. Trial advantage types: NT = neutral advantage, SA = single advantage, CA = congruent advantage, IA = incongruent advantage. mAdv = myopic advantage. -m and -f indicates the larger advantage. Error bars indicate bootstrapped 95% confidence limits.

Overall, participants made optimal initial path choices on 91.57% of the trials (excluding the NT and IA trials with equally optimal initial directions), and they did so quickly, with a group-average median planning time of 1.34 sec. As shown in Fig 3B, participants' selection of the initial path direction favored the optimal path across all of the advantage pairings, suggesting that they must be taking both constraints into account.

Both choice optimality and response times were influenced by the relative advantage of one path over another, by whether the optimal choice was based on a single advantage or a combination of two advantages, and by whether the larger advantage was the myopic, starting-point proximal advantage or the future, goal-proximal advantage (Fig 3B and 3C). To further characterize these effects, we present a set of planned comparisons taken from a mixed-effects regression model of trial-level initial path choices (with a probit linking function), with separate parameters for each trial advantage type and participant-level random intercepts. The model confirmed a significant difference in choice optimality among different advantage pairings shown in Fig 3B, $\chi^2(8) = 301.76$, $p < 0.001$. Response times (Fig 3C) also differed significantly among all 12 advantage pairings, $F(11, 1128) = 22.80$, $p < 0.001$, based on a linear model applied to the individual median z-scored first move response times. Detailed comparisons of the estimated marginal means (EMMs; with Bonferroni correction) based on both the choice model and the response time model (controlling for total path differences) confirmed that responses with an overall path advantage of 4 were faster and more often in the optimal direction than responses with a path advantage of 2 (adjusted $p$s < 0.01), and that responses in the IA trials were less optimal and slower than those in the SA and CA trials (adjusted $p$s < 0.01). Choice optimality rates in SA and CA trials were both near-ceiling and the difference between the associated response times was not statistically significant (adjusted $p$s > 0.1).

We now turn to the consideration of the relative influence of the myopic versus future advantages on speed and accuracy of initial path choices. Compared to the counterpart trials where the future advantage was larger (red points in Fig 3), trials where the myopic advantage was larger (blue points in Fig 3) had a significantly higher choice optimality rate (adjusted

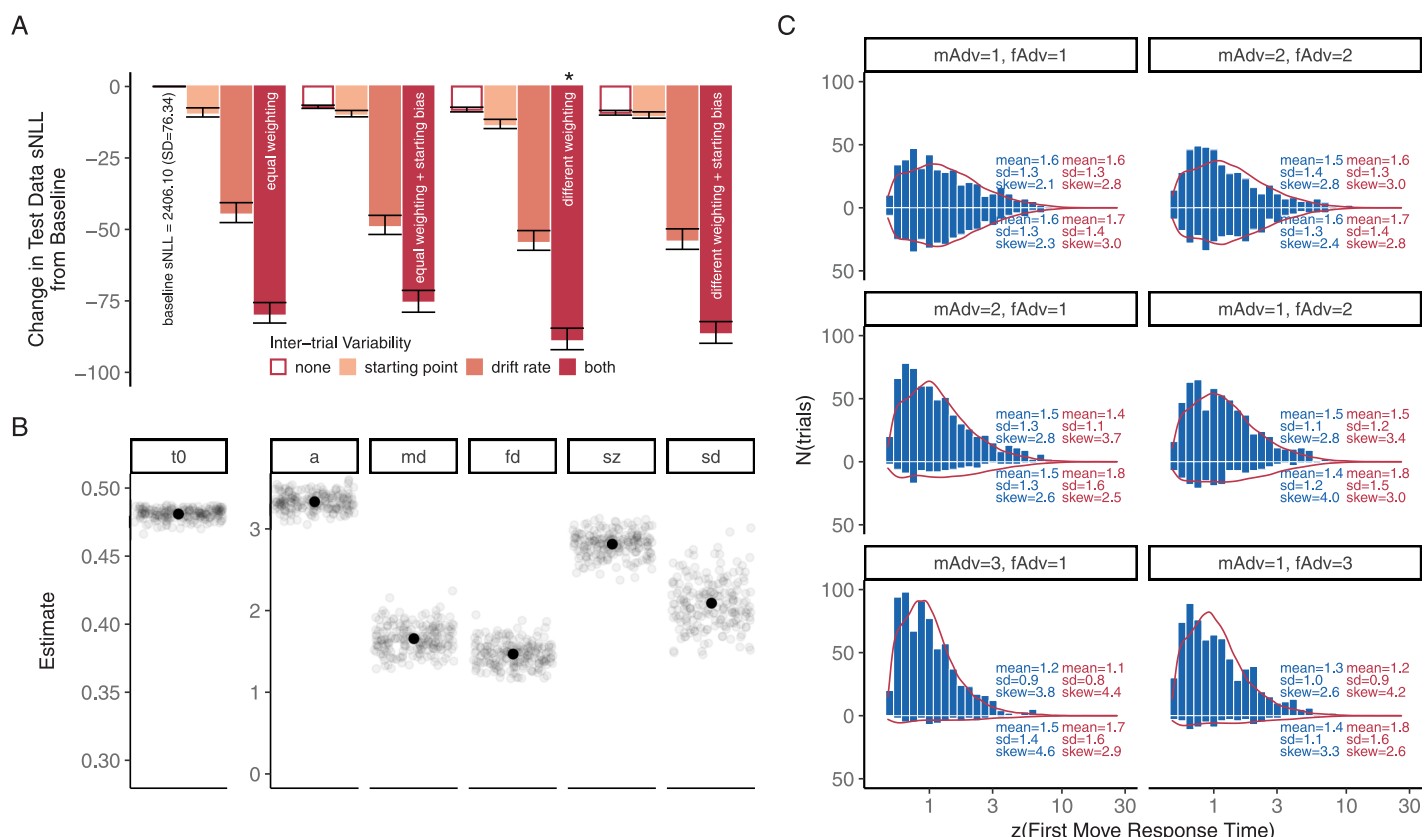

**Fig 4. Weighted integration of myopic and future constraints in initial-step decision making in the base task. A.** Test data objective (summed negative log-likelihood) of candidate drift-diffusion models, as marked on dark red bars, as compared to the baseline model (see text for details). Asterisk marks the winning model. **B.** Parameter estimates of the winning drift-diffusion model. $t_0$, non-decision time. $a$, decision bound. $md$ and $fd$, myopic and future advantage weights. $sz$, inter-trial variability of the starting point. $sd$, inter-trial variability of the drift rate. **C.** Predicted response time (RT) distributions (in red, sampled from the parameter estimates of the winning fit in the fold with the best test objective) and the empirical RT distributions (in blue). In C., the top two panels show the RT distributions associated with choices satisfying the myopic advantage on the top and those satisfying the future advantage on the bottom. The bottom four panels show the RT distributions for the correct responses on the top and error responses on the bottom. mAdv, myopic advantage. fAdv, future advantage. All error bars indicate 95% bootstrapped confidence limits.

$p<0.01$, EMM comparison from the choice model) and shorter response times (adjusted $p<0.05$, from the response time model) in IA and SA trials combined. An overall myopic bias is exhibited by a majority of the individual participants in the IA trials, though about 30% showed the opposite tendency when the two advantages are of equal magnitude.

Having observed the joint consideration of myopic and future advantages in deciding the first step as well as the myopic bias, we next turn to the comparison of drift-diffusion model variants in characterizing this decision process, focusing on the IA trials in which the two advantages lead to competing decision outcomes. This analysis suggested that the tendency to favor the path with the myopic advantage arose from a stronger weighting of the myopic advantage compared to the future advantage, and not from an initial bias favoring the myopic advantage (Fig 4A). The model with different weighting for the two advantages resulted in better fitting objective (summed negative log-likelihood) on the held-out test data across cross-validation folds, compared to the baseline model with equal weighting or the model with both equal weighting and a starting bias (adjusted $p$'s<0.001, pairwise EMM comparison based on a linear mixed-effects model of test objectives across all models, Bonferroni corrected). Accounting for additional starting bias led to worse fit in the equal weighting case (adjusted $p < 0.05$) and did not improve the fit in the different weighting case (adjusted $p > 0.5$). Models with

inter-trial variabilities in both the starting point and the drift rate also consistently outperformed those without (adjusted $p$'s$<0.001$).

The winning model estimated a ratio between the drift weights associated with the myopic and the future advantage at 1.13 (SD = 0.02, see Fig 4B). The model fitted the combined choice and response time data fairly well, though it under-predicted fast correct responses and over-predicted the occurrence of errors, particularly when the advantage difference was plus or minus one (Fig 4C, middle panel).

*Individual differences.* Considered together, participants in Experiment 1 made the optimal initial choices on over 91% of trials. However, the accuracy rate varied widely across participants, from 49.23%–100.0%. Participants also differed widely in how quickly they arrived at their first response (range of individual median response time: 0.60 sec–3.29 sec). To understand what factors differed between participants who made more and less accurate responses, we split the participant pool into two groups based on the median choice accuracy, with choice accuracy rates ranging from 94.44% to 100% for the higher accuracy group and from 49.23% to 94.37% for the lower accuracy group. Importantly, both groups showed a stronger influence of the myopic over the future advantage, though the effect was more prominent in reaction times for the more accurate group and more prominent in the probability of choosing the optimal path in the less accurate group (S1 Fig). Fitting the model separately to the higher and lower accuracy groups revealed that the more accurate group exhibited larger weights for both the myopic and future advantages and lower drift rate variability relative to the less accurate group (S2 Fig). In addition, we found that the more accurate group tended to weight the myopic and future constraints more nearly in accordance with an optimal, equal weighting of the two constraints. For the more accurate group, the ratio of the myopic weight relative to the future weight was estimated to be 1.09; for the less accurate group, the same ratio was estimated at 1.19. Compared to the distribution of differences obtained from 1000 pairs of randomly-split participant groups (see *Methods*), the observed difference in the ratio of the myopic and future weights between the two accuracy groups (-0.11) was close to the lower end of the 95% CI of the distribution of differences obtained from random splits [-0.14, 0.15], and only 66 of the 1000 random splits produced a more negative difference than the difference between the two groups. The observed difference is thus suggestive of, but not definitive evidence of, a reliable tendency for more accurate participants to weight the two constraints more nearly equally than less accurate participants.

## Experiment 2

Both the descriptive and the model fitting analyses from Experiment 1 indicated that people rely on both the relevant myopic and future constraints in selecting initial actions in the base task. Although participants were quite accurate overall in choosing the optimal path to the goal, they weighted the myopic constraint slightly more than the future constraint, and those who were less accurate tended to show a greater imbalance compared to the more accurate participants. We next investigated how people approach the same two-constraint problem when it appears as a subtask in a larger problem. Experiment 2 included the same set of trials used in Experiment 1 (*base trials*) and an additional set of trials where the base maze is the first part of a larger maze and leads to a subgoal that must be visited prior to reaching the final goal (*subgoal trials*; see Fig 2). In these subgoal trials, as illustrated in the figure, we varied the final goal location, to examine whether this factor might influence initial path choices, even though it is not relevant to the determination of the optimal path to the subgoal location.

We were interested in how the decision process behind the selection of the first step changes when people confront the more complex task as compared to when they solve the

base task. If participants adopted a hierarchical task decomposition approach, we may expect an increase in the initial non-decision time, followed by an identical decision process to find the shortest subgoal-reaching path compared to that in the base trials, with no influence of the position of the final goal on the choice of initial path direction. If participants instead extended the weighted consideration of multiple factors to the subgoal trials, we may expect no change in non-decision time, but instead changes in the weighting of the two relevant advantages along with some weighting of the irrelevant final goal in deciding on the initial path direction. This experiment also provides a further opportunity to consider differences in the constraint weightings between higher and lower accuracy groups, both with respect to the relative weighting of the myopic vs. the future advantage on initial path selection, and with respect to any observed weighting of the initially-irrelevant final goal position.

Experiment 2 was pre-registered through the Open Science Framework (https://osf.io/w78hu). The statistical tests, the drift-diffusion modeling, and the analyses of individual differences presented in this paper were developed after the experiment was completed and the pre-registered analyses were performed (we do not report the results of the pre-registered analyses as such since the reported analyses refine and extend them).

**Methods**

*Ethics statement*. This study was approved by the Stanford University Institutional Review Board under protocol No.7029. In the online experiment, participants were first shown a consent page and were instructed to continue to the study if they agree to participate or exit at any time if they decline to participate in any or all parts of the study.

*Participants*. We recruited 100 participants on Amazon Mechanical Turk with an identical set of eligibility criteria from Experiment 1, except that participants who previously participated in Experiment 1 were not eligible.

*Task design*. Participants completed two practice trials and 158 experimental trials in one session. Trial layouts were similar to those in Experiment 1, but the grid size was either 11×11 or 11×13 depending on the task condition (Fig 2). We added boundary walls to the grid canvas and removed the step count penalty for movements that resulted in wall collision. Participants were randomly assigned to one of the two orientation groups: group one (N = 50) received trials with left-to-right and right-to-left orientations, group two (N = 50) received trials with bottom-to-top and top-to-bottom orientations. Participants received a base completion compensation of $1.20 and a performance-based bonus on each trial ($0.03 for executing the shortest solution, $0.01 for a solution up to two steps longer than the shortest solution, and no bonus otherwise). The study took around 30 minutes and participants received an average of $5.63 for completing the study.

The experimental trials consisted of base trials (× 46), subgoal trials (× 92), multi-subgoal trials (× 16), and multi-subgoal control trials (× 4). In this paper, we report data from the base and the subgoal trials. The base trials contained the full set of 12 unique trial layouts used in Experiment 1 (see Table 1). The subgoal trials used the same set of advantage pairings but replaced the goal block in the base trials with a cell leading to a bottleneck (Fig 2). In the subgoal trials, the final goal block appeared on both ends of the goal column, as illustrated in the figure. Similar to Experiment 1, all trials were mirrored vertically except for the NT trials.

The multi-subgoal trials have two openings on the wall prior to the final goal which were structured so that the candidate paths through the two subgoals were equally optimal. On these trials, the start and the goal locations were fixed, but the subgoal locations varied and one of the subgoals was always closer to the goal. We also included the multi-subgoal control trials where we closed the subgoal further from the goal. The layout of these control trials did not overlap with any subgoal trials (see more information in the OSF repository).

*Exclusions.* All 100 participants successfully completed the experiment. At the trial level, we implemented the same set of exclusion criteria used in Experiment 1, excluding trials where participants took more than one minute to execute a first move (19 trials) or solved with five steps or more than the longer of the two main path candidates (ten trials), as well as trials with ill-identified initial path direction (180 trials). The exclusions resulted in a total of 13595 trial observations (98.5% of original dataset) for the reported analyses.

*Drift-diffusion modeling.* We tested whether, compared to the base trials, initial path choices and response times in the subgoal trials are accounted for by a different non-decision time or by changes to constraint weighting during the drift process. The baseline model capturing the decision process in the base trials was the winning model from Experiment 1, which includes six parameters ($t_0$, $a$, $md$, $fd$, $sz$, $sd$). We modeled a third addend to the trial drift rate in the subgoal trials, $gd$, representing the weight the final goal may carry during weighted information integration. $gd$ drives decision toward the upper decision bound (i.e., selecting the path that satisfies the myopic advantage) when the goal location is on the same direction with an initial move that satisfies the myopic advantage, and drives decision away from the upper decision bound otherwise. We also tested whether, and if so how, the advantage weights $md$ and $fd$ in the subgoal trials differed from those in the base trials, including shared additive change or independent changes to $md$ and $fd$. We subsequently added another variant, modeling a shared proportional change to both $md$ and $fd$. Response variables, model fitting, and model comparison were the same as in Experiment 1 (see Experiment 1 *Methods*), except that we used a 60/40 split for sampling the training data and the test data in the cross-validation folds.

*Analysis of individual differences.* Model fitting and parameter comparison for this analysis followed the same methods used for Experiment 1 (see Experiment 1 *Methods*), except that the median choice accuracy split resulted in N = 50 in each of the higher and lower accuracy groups, and we used a 35/15 train/test split for cross-validation in each group.

### Results and discussion

We compared how participants considered the various task factors when the same maze task either appeared alone (base trials) or as an initial subtask that leads to a subgoal (subgoal trials). In the subgoal trials, optimal initial paths are constrained by the same myopic and future path advantages as in the base trials, because final goal information is initially irrelevant. Before we turn to the drift-diffusion modeling results, we first review the overall response time and choice accuracy across the two task conditions.

Overall, performance degraded in the subgoal trials with both an accuracy cost and a response time cost. Participants selected the optimal initial direction more often in the base trials (mean optimal rate: 92.30%, range: 52.78%–100.0%) than in the subgoal trials (mean optimal rate: 88.03%, range: 51.43%–100.0%), $t(99) = 6.61$, $p < 0.001$. The group-average median response time of the first step in the subgoal trials (mean: 1.91 sec, range: 0.55 sec–9.45 sec) was about 0.40 sec longer than that in the base trials (mean: 1.53 sec, range: 0.58 sec–6.16 sec), $t(99) = 5.86$, $p < 0.001$. The accuracy and response time costs were seen across different trial advantage types (Fig 5).

The increased first move response times in the subgoal trials may reflect time taken to decompose the task and identify the final goal information as irrelevant to the subtask, but the simultaneous change in choice accuracy suggested that participants did not deploy the exact same decision process to solve the identical two-constraint task that is at core to selecting initial paths in both task conditions. We note that the benefit from congruent or single optimality-relevant constraints (as opposed to incongruent ones) was preserved, as path choices and response times showed similar patterns across the different trial advantage pairings in both

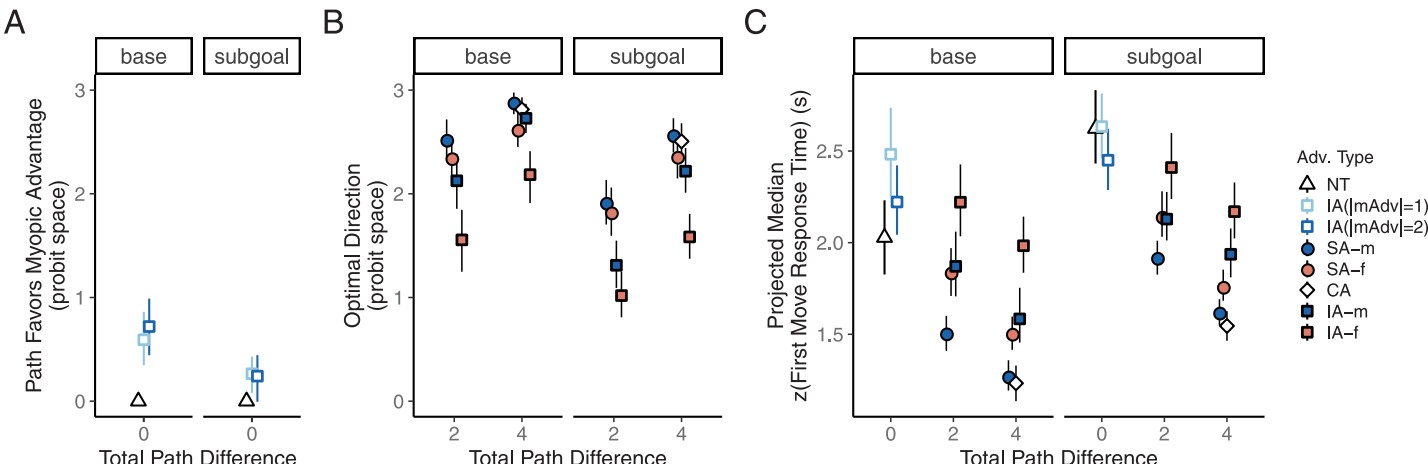

**Fig 5. Accuracy and time cost in initial path choices when the two-constraint maze is embedded as a subtask. A.** Path selection in the trials with equally optimal initial directions. As in Fig 3A, the value for NT trials was defined to be 0. **B.** Path selection across advantage pairings in base and subgoal trials. As in Fig 3B, the proportion of optimal trials for each individual was converted into a probit score, with individual probit scores larger than 3 or smaller than -3 capped at 3 or -3 before averaging. **C.** Response times in the base and subgoal trials. As in Fig 3C, the median zscore response times were projected back to the raw time scale in seconds.

task conditions (Fig 5B and 5C). This is confirmed by extending the path choice model and the median zscored response time model from Experiment 1 with task condition (base vs. subgoal) as a second predictor and accounting for interaction effects. We observed no significant interaction in either the choice model ($\chi^2(8) = 8.19$, $p = 0.42$, main-effects model as compared to an interaction model) or the response time model ($F(11, 2376) = 1.23$, $p = 0.26$).

Building on the winning model from Experiment 1 as a baseline, we next compared the decision process in the subgoal IA trials to that in the base IA trials. Model comparison suggested that changes in what and how different pieces of information are weighted in the decision process, but not an increase in initial non-decision time, accounted for the slowed and less accurate initial path choices in the subgoal trials. As shown in Fig 6B, adding a goal weight (*gd*, see Experiment 2 *Methods*) to the drift rate resulted in a significantly better model

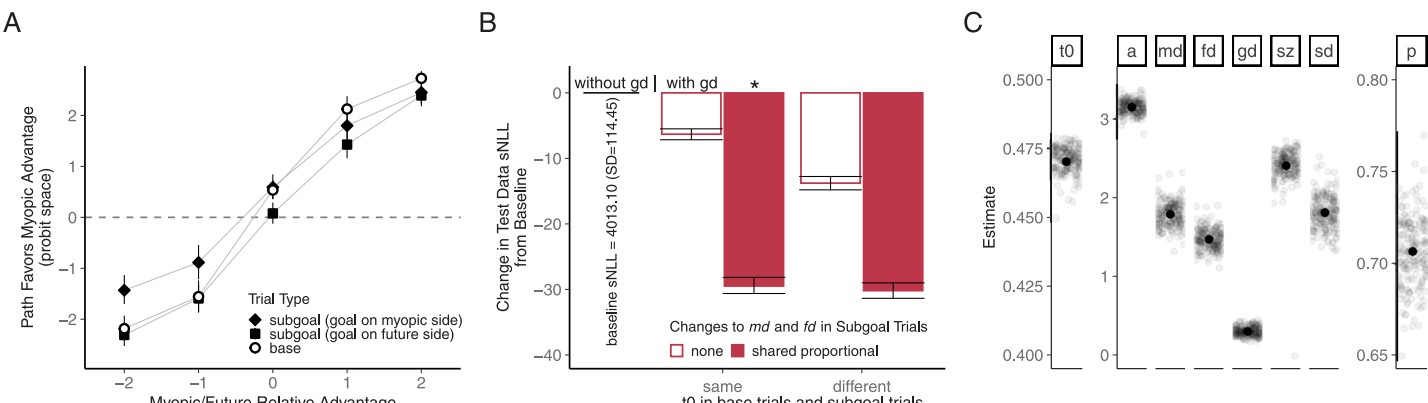

**Fig 6. Weighted consideration of subgoal-relevant and -irrelevant constraints approximated optimal choice behavior when the base task was embedded as an initial subgoal task. A.** Path choices in the IA trials. As in Figs 3B and 5B, the proportion of optimal trials for each individual was converted into a probit score, with individual probit scores larger than 3 or smaller than -3 capped at 3 or -3 before averaging. **B.** Test data objective (summed negative log-likelihood) of the candidate drift-diffusion models compared to the baseline model. Asterisk marks the winning model. For empirical and predicted response time distributions, see S4 Fig. **C.** Parameter estimates from the winning model. $t_0$, non-decision time. *a*, decision bound. *md* and *fd*, myopic and future advantage weights. *gd*, goal weight. *sz*, inter-trial variability of the starting point. *sd*, inter-trial variability of the drift rate. *p*, proportional change to *md* and *fd* in the subgoal trials.

compared to the baseline model (adjusted $p < 0.001$, pairwise EMM comparisons based on a linear mixed-effects model fit to the test objectives from all model variants, Bonferroni corrected). This echos the pattern of the initial path choices in the IA trials, which clearly showed a biasing influence from the irrelevant final goal (Fig 6A).

We also found that a proportional decrease (modeled by a weight multiplier $p$) in the weights of the optimality-relevant path advantages (*md* and *fd*) further accounted for the decreased choice optimality and slowed response times in the subgoal trials (Fig 6B). This echoed the similarity in the response patterns across advantage pairings shown in Fig 5, suggesting an overall degradation in the weighting of the two relevant constraints in the more complex task. The proportional weight change model was better than the model with an equal decrement in *md* and *fd* (adjusted $p < 0.001$), and on par with the independent weight change model (adjusted $p > 0.9$) with one less free parameter. Moreover, modeling a separate non-decision time in the subgoal trials for the proportional weight change model did not lead to significantly better test data likelihood (adjusted $p = 0.13$), indicating that the process of deciding between the path choices could initiate after about equal time in both task conditions, as the slowed response time was accounted for by reduced weighting of the relevant constraints.

The winning model estimated that the weight placed on the irrelevant final goal (*gd*) was about 0.31 (SD = 0.06; Fig 6C). It is a much smaller weight compared to the weights associated with the optimality-relevant myopic and future advantages in the subgoal trials, which were estimated at *md* = 1.27 (SD = 0.12), *fd* = 1.04 (SD = 0.10) after about a 30% (SD = 2%) decrease compared to their values in the base trials, with their ratio preserved at 1.22 (SD = 0.03).

**Individual differences**. As in Experiment 1, there were large individual differences in participants' overall accuracy (S3 Fig). As before, the qualitative pattern of effects exhibited in the combined results across all participants was also exhibited by participants in both the higher accuracy group (accuracy rate: 91.67%–100.0%) and the lower accuracy group (accuracy rate: 53.85%–91.59%). When fitting the model separately to data from the group with overall higher accuracy and the group with overall lower accuracy, the lower accuracy group showed a higher degree of drift-rate variability (S5 Fig), consistent with that found in Experiment 1.

The groups also differed in the relative weighting of the different task factors, with the more accurate group again showing a tendency to place more nearly optimal (i.e., equal) weighting of the initially-relevant myopic and future constraints (Fig 7C). For the more accurate group, the ratio of myopic to future weights was 1.17, compared to 1.30 for the less accurate group. The difference between the weight ratio estimated from the full split-group fits (-0.14) was

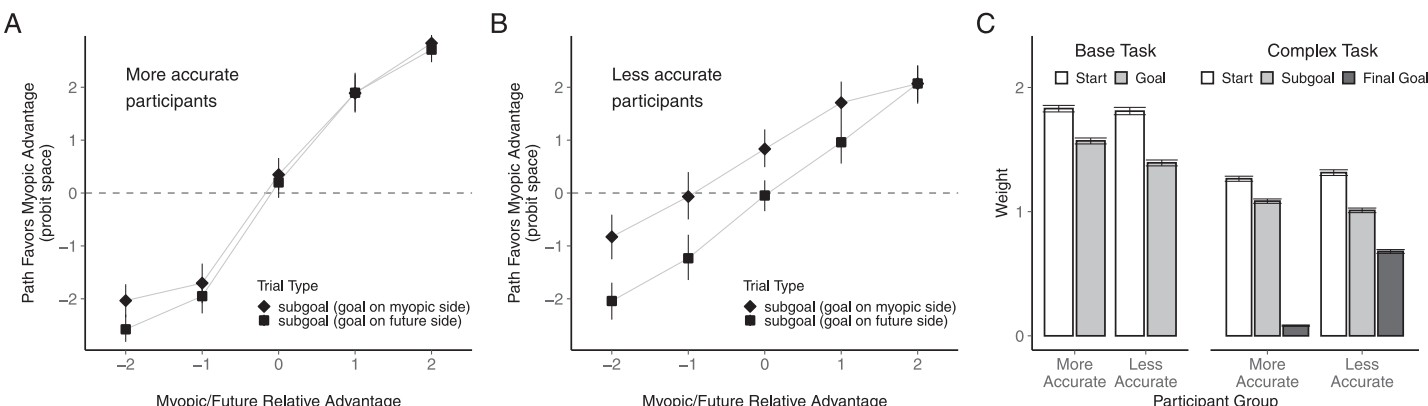

**Fig 7. Individual differences in weighting subgoal-relevant factors and the subgoal-irrelevant final goal. A.** and **B.** Subgoal trial path choices associated with the two accuracy groups. As in Fig 6A, the proportion of optimal trials for each individual was converted into a probit score, with individual probit scores larger than 3 or smaller than -3 capped at 3 or -3 before averaging. **C.** Drift weight estimates in both task conditions for the two accuracy groups.

again very close to the lower end of the baseline difference distribution obtained from 1000 pairs of randomly-split participant groups: the 95% CI spanned the interval [-0.15, 0.16] and only 36 of the 1000 random splits produced a more negative difference. The probability that the difference between the high and low accuracy groups would have been as far down in the negative tail of the baseline distribution as it was in both experiments is very low (the joint probability of the two events occurring by chance under random splits is $.066 \times .036 = .0024$).

More strikingly, the influence of the final goal on initial path choices in the subgoal trials is clearly much smaller for the more accurate participants (Fig 7A) than it is for the less accurate participants (Fig 7B). This is reflected in the markedly different goal drift weights estimated in the DDM model fits of the two accuracy groups (Fig 7C). For the group with higher overall accuracy, the goal carried a reliable, but very small weight (0.08). For the group with lower overall accuracy, however, the goal weight was estimated at 0.68 (difference from full split-group estimates: -0.58, 95% CI of the baseline distribution: [-0.30, 0.29]). Both groups showed around 30% degradation in the weighting of the optimality-relevant myopic and future advantages in the complex task (more accurate group: 30.89%; less accurate group: 27.49%; difference from full split-group estimates: -3%, 95% CI of the baseline distribution: [-10%, 10%]).

## General discussion

In this work, we have proposed a weighted constraint satisfaction approach to goal-directed decision making, contrasting this with a hierarchical task decomposition approach. If participants were strictly optimal, task-decomposing planners, we would expect that their decision processes in the complex maze task would correspond to finding the best subgoal-reaching initial path and thus resemble the choices made in the base maze task. Instead, we found that participants' initial path choices were less likely to be optimal in the complex task, and that their choices were influenced by both the subgoal-relevant constraints and a subgoal-irrelevant factor (the location of the final goal).

We used a drift-diffusion modeling (DDM) framework to jointly account for path choices and response times in our experiments. The results of the modeling effort revealed several things. First, when confronted with the base task, participants' decisions were influenced by both the relevant starting-point proximal (myopic) constraint and the goal-proximal (future) constraint, but to differing degrees. This was accounted for in the drift-diffusion model by assigning a greater weight for the myopic over the future constraint in determining the overall drift direction of the diffusion process. In the more complex task, the weighting of the same two optimality-relevant constraints was reduced, and participants' weighting of the final goal was strong enough to lead to suboptimal initial step choices on some trials. Responses were overall slower in the more complex task, a finding that, taken by itself, might seem to reflect the time required to focus in on the embedded subgoal in a hierarchical task decomposition process. However, changes in the degree to which the optimality-relevant or -irrelevant task factors were weighted, rather than a simple additive offset on the total time to make the decision, accounted simultaneously for the reduction in choice accuracy and for the slowed responses in the more complex task.

While participants' responses usually favored the optimal path, they did not always do so, and there were individual differences in how closely participants approximated optimal choice behavior in our tasks. The drift-diffusion model implementation of our weighted constraint satisfaction account can exhibit varying degrees of optimality in the predicted initial path selection, allowing us to capture these variations across participants. When the weights assigned to optimality-relevant constraints are strong compared to the magnitude of sources of variability, and when the decision criterion is chosen sufficiently conservatively, the optimal

path will be selected by a drift-diffusion process with a probability approaching 1. We explored what factors differed between participants who were more or less accurate in selecting the optimal path and found that more accurate participants showed less trial-to-trial variability in the overall drift direction and placed greater weight on the relevant constraints than less accurate participants. These differences were also accompanied by differences in how well the participants conformed to the ideal pattern of constraint weighting in our experiments (Fig 7C). More accurate participants placed much less weight on the irrelevant final goal compared to the less accurate participants, and also tended to exhibit more balanced weighting of the relevant myopic and future constraints. Thus, greater approximation to optimality was not only associated with a stronger overall signal-to-noise profile in the decision process, but also with a greater degree of optimality in weighting the different constraints.

## Weighted constraint satisfaction and optimization

Taken together, these findings support the view that goal-directed planning can usefully be viewed as a weighted constraint satisfaction process, at least within the current task setting, and that optimizing planning can be viewed as occurring through adjusting the weighting of constraints toward values that favor ones relevant to selecting the optimal action. We do not intend to suggest that explicit task decomposition never occurs. However, even when it does, it may still be the case that information not strictly necessary within a subtask context contributes a graded influence within the subtask context. In this context, it is important that even participants who made overall highly accurate choices showed some imbalance in their weighting of the myopic and future constraints and placed some weight on the irrelevant final goal. This consistency across groups of higher overall accuracy and lower overall accuracy is consistent with the idea that all participants relied on the same constraint weighting mechanism, but that participants in the more accurate group modulated their constraint weightings more successfully.

A weighted consideration of multiple aspects of a situation, especially including some that are irrelevant, can sometimes lead to suboptimal decision outcomes in the context of a particular task. Such a tendency, however, might be viewed as a consequence of a more global, perhaps evolutionary, adaptation. Studies have separately suggested that optimal task decomposition and decision suboptimality can both result from optimizing meta-level costs associated with representation and computation [7–9, 11, 25, 26]. One of the motivations for the idea that humans might consistently adopt a weighted constraint satisfaction approach stems from a similar consideration. The computational cost associated with switching between parallel or hierarchical planning processes when facing different situations may in fact be higher compared to the consistent use of parallel processing of multiple exploitable factors, which would result in approximately optimal behavior as long as information is appropriately weighted. The simultaneous consideration of multiple pieces of potentially relevant information can also be useful in problems where an exact hierarchical decomposition may not be possible, and also support the flexible incorporation of new information as it arises in a dynamically-changing environment. The myopic bias that is suboptimal in our task context may also reflect a rational allocation of attention to the greater certainty of the near than the distant future or the tendency to plan only partially as a result of sensitivity to meta-level planning costs [26, 27]. The suboptimal consideration of the irrelevant final goal can also be seen as rational or at least natural for its possible value, considering that future information can often be relevant to the choices we make in the early stages of complex task situations, such as the packing stage of planning for a long trip, as mentioned in the *Introduction*.

Our results echo the simultaneous consideration of immediate and future constraints found in other cognitive domains (e.g., typing and speech production), as similarly revealed

through both optimal and suboptimal aspects of participants' behavior when planning and executing a sequence of actions [16, 28]. The models used to account for this behavior generally weight considerations relevant to the immediate next action most heavily, with successively less weight to items more remote in the sequence. Indeed, we may understand at least the myopic bias we have observed in our participants' behavior in terms of such a general tendency, in that we place the largest weight on the constraints most proximal to the immediate action at hand (accounting for the myopic bias) while allocating successively decreasing weight to future constraints (accounting for the reduced weight of the relevant future constraint and even less weight of the irrelevant final goal). Optimizing goal-directed planning can thus be thought of as eliciting an adjustment from a default or baseline future-discounting weighting toward the optimal weighting values in a given task setting.

## Cognitive control as modulation of constraint weightings

A long history of research in the fields of attention and cognitive control has argued for the view that cognitive control occurs through the modulation of the weight placed on relevant vs. irrelevant factors [29–31]. The details of how such weighting is implemented have varied, but a common thread in much of this work is that strong, relatively automatic behavioral tendencies may often be sufficient to explain aspects of routine action, such as reading and typing of relatively familiar words or taking the same route to work that one has taken over a period of years. When current task demands require modification of these tendencies, these approaches argue that we do not simply replace the habitual mechanisms with others, but that we modulate these mechanisms, biasing their operation so that the relevant factors are more dominant. This approach provides a natural way of accounting for the fact that habitual tendencies (e.g., to produce the utterance "red" when looking at the word RED) still appear to influence processing even when participants are instructed to do something that may be less habitual (e.g., name the color of the ink used to print the word; [29]). Indeed, it has been argued that the ability to exert such control is a key mechanism underlying the positive correlation in task performance across a very wide range of cognitive tasks, including so-called fluid or culture-fair intelligence tests thought to measure the general intelligence factor $g$ [32]. In this light, our finding that more accurate participants showed more optimal weighting than less accurate participants could be interpreted as indicating that they exerted a greater degree of task-relevant control than less accurate participants.

## Visual factors and shifts of attention

It is worth noting that some aspects of our findings may depend at least in part on particular features of the experimental design, including the fact that information about the starting point, final goal, and subgoal in the complex task condition of Experiment 2 is visually available throughout the experiment. One possibility is that the visual salience of the final goal may have strengthened its biasing influence in the complex task. While our experiment does not rule this out, the extent of any such effect may be modulated by other specific aspects of our experiments. The same visual marker is used to indicate the optimality-relevant goal in the base task, possibly increasing participants' weighting of the irrelevant final goal in the complex task as a carry-over effect. In this context, though, it is worth pointing once again at the fact that more accurate participants were far better at reducing the biasing effect of the optimality-irrelevant goal cue, indicating that, like other constraints operating on the weightings people place on various factors, the biasing effect of this cue is not a simple invariant effect of visual distraction but is subject to cognitive control. It is also worth noting that the fact that the two tasks were interleaved may have encouraged the realization that the same two constraints were

the only ones relevant to selecting the initial step, potentially contributing to the equal non-decision time cost found across both tasks. Future work would be needed to fully disentangle these different contributing factors.

Given the visual availability of all of the constraining influences, the time course and outcomes that are captured by the drift-diffusion model could arise from a process that involves several alternations of constraint weightings, possibly associated with changes in eye fixation. Some models of preferential choice include such alternations [33, 34], and eye movement data provides evidence that such alternations do occur in preferential choices between visually presented options [20]. Viewed in this light, the overall weightings we obtained in our model fits could reflect in part the fraction of the decision time devoted to attending to one or the other sources of constraint, with the variability parameters of the DDM approximately capturing the variability in the selection and timing of attentional shifts. In this context, it is interesting to note that the weightings did not seem to shift in an all-or-none fashion with shifts in eye position in the study of preferential choice between alternatives [20]. Instead, the weighting of the fixated choice alternative was enhanced relative to the weighting of the other alternative, so that the weightings changed in a graded fashion as the eyes moved. In future work, it would be possible to investigate whether a similar effect would be observed in planning situations such as the one we have explored in our experiments. In any case, it is worth emphasising that, even if there is attention shifting underlying the combined influences of different sources of information, the overall outcome of the decision process still reflects the weighted influences of multiple constraining factors.

## Extensions, related models, and future directions

An important future step will be to examine whether a weighted constraint satisfaction mechanism also supports more complex human problem solving beyond the navigational or other everyday tasks we have considered up to this point, for example, in tasks that require a larger number of intermediate steps or more abstract forms of reasoning. One such example is mathematical theorem proving, in which successful reasoning depends on constraining search for a proof based both on the givens in a problem and the statement-to-be-proven at the end of the proof sequence [35]. We may therefore expect a weighted constraint satisfaction mechanism to also underlie more advanced forms of reasoning, with immediate and long-range information (e.g., known conditions and the goal statement) being simultaneously processed to facilitate finding important intermediate steps, and a strengthened weighting of a subset of the constraints marking a temporarily enhanced focus on an intermediate subtask (e.g., proving a lemma).

We have focused on understanding the weighting of different constraints leading to the very first action, but our account naturally extends to multi-step decision making over time. In the more complex task, the initially-irrelevant goal information would eventually constrain the path after the bottleneck location. In general, generating consecutive steps in this task may be supported by a weighting shift as decision making progresses, so that, for example, as a person proceeds to and through the bottleneck leading to the passageway containing the goal, they will place greater weight on the final goal to support subsequent action selection. Investigating how constraint weightings are modulated over time is another exciting future direction.

The drift-diffusion framework has illuminated human cognition across multiple domains, including perception, memory, and decision making [1, 3, 20–22]. In our tasks, it has provided a useful characterization of the constraint weighting dynamics of the underlying decision process. Existing work on planning and hierarchical computation in a variety of tasks has leveraged probabilistic models or the reinforcement learning framework to model the hierarchical task decomposition approach [6, 8, 10, 36]. An important direction for future work will be to explore how these alternative approaches to modeling goal-directed planning inter-relate. It is

possible to envision, for example, that adjustments to constraint weights are driven by a process like reinforcement learning, or, as suggested above, that the path choices can be viewed as Bayes-optimal when resource constraints are also taken into account.

One particularly exciting direction for our own future investigations will be the exploration of contemporary attention-based neural network architectures as a mechanistic instantiation of the process of assigning and using weightings of various factors in a decision making context [37, 38]. These architectures have been used for machine translation, in which weighted attention to relevant words in a source sentence in one language is used to guide choices of words to produce in the translation [38]. Such models adapt their attention weights for each successive item in an output sequence and acquire sensitivity to latent hierarchical structures [39]. We are excited by the prospect of adapting such models to the goal of understanding how humans generate sequential goal-directed actions and acquire sensitivity to latent hierarchical structures in a wide range of task contexts and by the prospect of the potential convergence between such an approach and other frameworks [40].

## Conclusion

In summary, we propose that goal-directed decision making can be characterized as a parallel, weighted constraint satisfaction process. In our experiments, simultaneous and appropriately-weighted consideration of multiple constraints helped account for the detailed choice and response time patterns from human participants, both when a simple maze navigation task was presented alone and when it was embedded in a larger maze. In both task settings, participants adapted their weighting of the relevant myopic and future constraints in the direction of optimizing their initial path choices, but deviations from optimality revealed gradations in the weighting of optimality-relevant constraints and revealed suboptimal consideration of optimality-irrelevant factors which biased decision outcomes. These results suggest that human choice behavior can be usefully viewed as a context-sensitive modulation of the weighting of multiple constraints, which can both exploit multiple pieces of information relevant to decision making and produce an approximation to the optimal choice outcomes predicted by a hierarchical task decomposition process.

## Supporting information

**S1 Text. Supplementary methods.** Use of additional fitting runs to avoid local minima. (PDF)

**S1 Fig. Experiment 1 path choices and response times for groups of higher and lower accuracy.** Visualization and notation as in Fig 3. The individual median zscore response times were projected to the raw time scale in seconds using the subgroup-average of mean response time and standard deviation. (PDF)

**S2 Fig. Experiment 1 IA trial responses and drift-diffusion model fits among groups of higher and lower accuracy.** Visualization and notation as in Fig 4B and 4C. (PDF)

**S3 Fig. Experiment 2 path choices and response times for groups of higher and lower accuracy.** Visualization as in Fig 5. The individual median zscore response times were projected to the raw time scale in seconds using the subgroup-average of mean response time and standard deviation. (PDF)

**S4 Fig. Experiment 2 model-predicted response times (in red) and empirical response times (in blue).** **A**. Base trials. **B**. Subgoal trials. Visualization and notations as in Fig 4C. (PDF)

**S5 Fig. Experiment 2 IA trial responses and drift-diffusion model fits for groups of higher and lower accuracy.** Left panel, model parameter estimates. Middle panel, base trials. Right panel, subgoal trials. Visualization and notations as in Fig 6C and S4 Fig. (PDF)

**S6 Fig. Experiment 2 survey responses.** Participants' self-reported considerations of different task elements was in line with the behavioral analyses presented above. In free-form responses, when asked how they came up with the route to reach the goal, their responses consistently suggested that the reasoning process was rapid and intuitive. Likert scale: 1=never, 2=rarely, 3=sometimes, 4=often, 5=always. Sprite denotes the movable blue block. Error bars indicate bootstrapped 95% confidence limits. (PDF)

## Acknowledgments

We thank Stanford University for supporting the effort of both authors on this project, and Tobias Gerstenberg and the members of the second author's lab group for useful discussions.

## Author Contributions

**Conceptualization:** Yuxuan Li, James L. McClelland.

**Data curation:** Yuxuan Li.

**Formal analysis:** Yuxuan Li, James L. McClelland.

**Investigation:** Yuxuan Li.

**Methodology:** Yuxuan Li, James L. McClelland.

**Supervision:** James L. McClelland.

**Visualization:** Yuxuan Li, James L. McClelland.

**Writing – original draft:** Yuxuan Li.

**Writing – review & editing:** Yuxuan Li, James L. McClelland.

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
