## [Decision Letter · Decision Letter 0]

6 Dec 2021

Dear Ms. Li,

Thank you very much for submitting your manuscript "A weighted constraint satisfaction approach to human goal-directed decision making" for consideration at PLOS Computational Biology.

As with all papers reviewed by the journal, your manuscript was reviewed by members of the editorial board and by several independent reviewers. In light of the reviews (below this email), we would like to invite the resubmission of a significantly-revised version that takes into account the reviewers' comments.

Three experts have reviewed your manuscript and all of them have concerns that need addressing, albeit to differing extents. In my own reading, a major question is why and how the drift diffusion model was chosen in this context, and how it was integrated with the planning literature used as a framework. On top of that, as brought forward by all reviewers, many experimental and modeling details need to be clarified. When this is done, I think it could be an interesting addition to the literature.

We cannot make any decision about publication until we have seen the revised manuscript and your response to the reviewers' comments. Your revised manuscript is also likely to be sent to reviewers for further evaluation.

Sincerely,

Marieke Karlijn van Vugt, PhD

Associate Editor

PLOS Computational Biology

Natalia Komarova

Deputy Editor

PLOS Computational Biology

Three experts have reviewed your manuscript and all of them have concerns that need addressing, albeit to differing extents. In my own reading, a major question is why and how the drift diffusion model was chosen in this context, and how it was integrated with the planning literature used as a framework. On top of that, as brought forward by all reviewers, many experimental and modeling details need to be clarified. When this is done, I think it could be an interesting addition to the literature.

Reviewer's Responses to Questions

**Comments to the Authors:**

Reviewer #1: Li & McClelland present two experiments combined with drift-diffusion modeling asking how people weigh distal and proximal constraints while navigating towards a goal. The problem is interesting and timely, but I am afraid the paper suffers from a number of quite severe issues.

First, unfortunately I found most of the exposition quite incomprehensible despite being well versed in both drift-diffusion modeling and reinforcement learning (which although not used here directly, the authors use for motivation). This was due two broad categories of issues: 1) many ideas and terms specific to the project were used matter-of-factly before they were introduced or adequately explained, and 2) some details were missing entirely. For example, it was never entirely clear to me how the weighting is actually implemented, which is a core topic of the paper. From the plots I could infer that it was based on the drift rate, but what kind of function of the drift rate? Is it just a separate single drift rate estimated for each condition? If so, how does the distal goal come into play in Exp 2? The full modeling assumptions are unclear. Mixed-effects are mentioned, but the details are scattered and hard to piece together. One sentence of the methods reads "The walls on the filler trials were of length- 3, 5, or 7 and were sometimes shifted from the center..". When is "sometimes"? What is the algorithm for designing the state space? I thought carefully about how to phrase the following, but I think it's fair to say the paper needs to be re-worked essentially from scratch, carefully including all details and slowly building up the questions and their answers.

Second, related and more importantly, the specific scientific questions being asked were themselves not altogether clear, and may represent a strawman. For one, I wasn't sure whether a claim was being made about participants using a hierarchical representation for this particular task. The introduction suggests thinking about these problems in terms of hierarchical reinforcement learning, and as far as I could tell, the idea being that people might select an abstract sub-goal but then still be distracted by high-level features of the state space (the ultimate goal) even after they "step down" one level to deal with the subgoal. But, there is no evidence presented that people use a hierarchical representation in this task at all. If they do, it is also important to understand *what* representation is being used. There is no reason to think that the only subgoal location would be at the door. For example, participants may create subgoals directly on each side of a barrier. A question then is whether it is the *decomposition* that is interfering with decision making (i.e. maybe they chose a poor decomposition) rather than (or in addition to) at an algorithmic level whether participants reason across or within modules of the decomposition appropriately. Putting hierarchy aside, one may interpret the paper as asking about how people weigh proximal vs distal constraints, but from this perspective the results appear to be entirely unsurprising: people are worse at the harder problem in Experiment 2 and have a myopic bias. These results in themselves unfortunately do not add anything new.

Reviewer #2: The paper “A weighted constraint satisfaction approach to human goal-directed decision making” by authors Li & McClelland investigates how reaction time and accuracy change when participants navigate to a sub-goal in a maze (the “sub-goal” condition) compared to when this sub-goal was not embedded in a higher-level structure and the only goal in the task (the “base” condition). The key comparison relates to the influence of path “advantages” where it was easier either to turn around an obstacle from the start, at the goal, or both. The results using drift diffusion modelling show that while participants assign high weight to these advantages in the base condition, seemingly trading off these two components, their performance is slightly impaired in the subgoal condition, albeit with similar relative weighting between advantages. The paper conclude that this provides evidence for a parallel consideration of planning constraints.

The paper is well written and presents a clever paradigm in which drift diffusion modelling can illuminate planning behaviour. While I would be happy for this paper to be published in its current form, I do have some concerns relating to interpretation and terminology and would appreciate it if these were addressed or clarified in a revised manuscript. But do pick and choose from these as you please.

Major:

1. I’m not sure the results directly map onto the sequential/parallel distinction made in Fig. 1. It is unclear to me how the finding that the goal in experiment 2 influences myopic advantage relates to parallel planning. Wouldn’t a claim that this shows “forward” planning rather than “backward” (from the goal) be equally supported without requiring this additional claim?

2. I also wonder in how far this “myopic” bias in the sub-goal condition is favoured by the goal being clearly visible, whereas planning to the sub-goal (and working out its advantage) seems more difficult when the sub-goal is only indirectly marked using the bottleneck next to it. My hunch would be that if the sub-goal were marked, planning would look less affected by the irrelevant goal position. In other words, is this mainly an effect of difficulty because the sub-goal needs to be inferred? This is particularly important as trials were interleaved as far as I understand it. Are there perhaps enough consecutive sub-goal trials to check if participants improved within such a mini-sequence?

Minor:

3. I have some quibbles about the “advantage” and “constraint-satisfaction” terminology. I see that advantage favours either forward or backward planning in the base condition, but I found this term unintuitive at first, because for the purpose of the task there is no advantage at all, just a biasing influence to take an easy way around the obstacle (and thus possibly make it easier to find the optimal route). Perhaps a term like “salience” could be more appropriate?

4. Similarly, my understanding is that constraints often mean something a bit different in the literature, e.g. budget constraints in economics, or time-constraints in navigation, for example in a similar setting in (Ringstrom and Schrater, 2019). In the present study, the constraints are essentially the maze layout (boundaries and bottlenecks), which are seldom referred to as constraints in other studies. Nevertheless, I agree that the distinction between having to overcome the barriers near the start and the goal is interesting and relevant. However, I would see this more in terms of “chunking”, or “options” within the Reinforcement Learning framework (Precup et al., 1998) rather than constraints. And in my personal view only the multi-subgoal task fits the more conventional sense of a constraint. What was the reasoning for not including this trial-type in the analyses?

5. I found Figs 3 and 5 to be quite busy and had trouble working out what the relevant effects were without looking at the text in detail. Perhaps adding a key statistical test with significance to the plots would help. E.g. the IA-CA comparison referred to in line 223.

6. I find the claim (lines 441f) that the model does not need to separate problem decomposition and execution a little unconvincing. In the current version the constraints are hard-coded and so the model really only speaks to execution over a pre-decomposed space. Granted I can’t think of a better decomposition for this current task, but any modification of the boundaries would always entail a necessary revision of the components of the model.

References:

Precup, D., Sutton, R., and Singh, S. (1998). Theoretical results on reinforcement learning with temporally abstract options. Machine Learning: ECML-98. Springer Berlin Heidelberg 382–393.

Ringstrom, T.J., and Schrater, P.R. (2019). Constraint Satisfaction Propagation: Non-stationary Policy Synthesis for Temporal Logic Planning. ArXiv.

Reviewer #3: Summary:

This paper reports two experiments and modeling that test the hypothesis that hierarchical planning involves parallel processing of information about longer-term goals even when people are pursuing a sub goal. This claim is contrasted with the idea that processing of sub goals and goals occur sequentially and separately. Exp 1 examines RTs when people navigate simple mazes where they must navigate around two barriers to reach a goal. Start and goal states were placed such that rounding the first barrier from the start or rounding the second barrier to reach the goal were longer or shorter on different trials. By examining reaction times on different trials, this design allows for assessing differential biases to plan based on the configuration around the start versus goal locations. Results for Exp 1 show that people use both start and goal configuration information to plan (as evidenced by the general tendency to take the shortest path that integrates information on both the start and goal sides), but that people have a small bias towards the start configuration. Exp 2 examines RTs when Exp 1 mazes are embedded in larger mazes, such that the original maze is a subtask for completing the new maze. Using differentially parameterized drift diffusion models, they find that the presence of a final goal that should not affect optimal behavior or processing on the subtask influences first-move RTs. Since reaching the final goal is a supertask and reaching the other side of the first room is a subtask, they conclude that this influence indicates that goal processing occurs in parallel with planning to the subgoals.

Main Review:

I enjoyed reading this paper and appreciate the experimental approach taken. The design of the base mazes versus embedded mazes was especially clever and intuitive. However, I have several major concerns about confounds in the experimental design, the space of models tested, and the overall framing of the paper.

1. There seem to be confounds in Exp 2 that need to be controlled for. Importantly, the base and subgoal mazes are not only different tasks, they also differ perceptually (e.g., their complexity and the objects present). The authors' main claim regards how planning in a subtask context is influenced by a supertask context and *not* by just the perceptual differences, but these are both present. Another way to think about it is this: the presence of the red goal block on the subgoal trials (Fig 2) could have influenced subtask processing because it is being maintained as *as a goal in a hierarchy*, or simply because it *perceptually distracting*. From what I understand, the results of Exp. 2 can only support their main argument if the former and not the latter is true, and so this needs to be ruled out. I could not tell if the multiple subgoal trials were introduced to account for this, and the authors note that these data were not analyzed (line 305). Would some analysis of these trials be able to account for the perceptual effect of the red goal? If not, then ideally, the authors could run controls where the base/subgoal trials are perceptually equivalent. One possibility is that participants could be shown the same mazes but told before viewing the maze that they need to get to the star (subgoal version) or just the door (base version). Other variations are also possible.

2. Another issue involves the space of models tested---in particular, the assumption of hierarchical planning occurring in the task. For Exp 2., the authors modeled their data with drift diffusion models where the drift rate included parameters associated with the start location (the myopic advantage weight), the subgoal location (the future advantage weight), and the goal location (the final goal weight). It sounds like they only fit the goal weight after fitting and then freezing the myopic and future weights (line 396), which roughly corresponds to fixing the parameters of the subtask context and then testing if the supertask context explains additional variance. However, these analyses all presuppose hierarchy---what if people are not planning hierarchically at all and are simply engaged in flat planning? In the DDM modeling framework they use, this corresponds to fitting all three parameters simultaneously (I think). The authors should compare the fit of this model to the fit of the ones already tested (e.g., by reporting delta AIC).

3. Additionally, while I appreciate the authors' attempts to determine whether subtask processing is isolated from a full hierarchical context, couldn't interference from the goal just mean people are doing flat planning? Put another way, when people are flat planning, they are planning in a context that includes the entire task (all the constraints, the start, bottleneck, and goal). Other algorithmic considerations could come into play within a single flat planning context---as a concrete example, people could be performing heuristic search using a start-to-goal distance heuristic that biases their initial action or reaction times. I understand that a general challenge with modeling these kinds of tasks is that the space of possible planning algorithms is large. Nonetheless, the authors want to make a fairly strong claim about a class of planning algorithms (ones that involve hierarchical decomposition) but they need to provide evidence that this type of algorithm is being used by participants to make such a claim.

4. I found the contrast between serial and parallel processing during hierarchical planning to pose somewhat of a false dichotomy. Even if people are hierarchically decomposing a task, couldn't interference from the goal context be because of switching between goal/subgoal processing (i.e., multiplexing)?

Minor points

- At several points, the authors claim that parallel processing is "the optimal approach" (e.g., line 68). It is not clear what this means---optimal with respect to what?

- Figure 3 - why use probit transform? couldn't you just show the probabilities?

- Lines 315-317 - the multigoal trials seem critical but I'm not sure what these sentences are saying.

- It seems like a DDM model was used in this work because it can account for reaction times, however, work on planning and hierarchical planning is generally done with markov decision processes (MDPs; e.g., the optimal behavioral hierarchy work by Solway et al., 2015, other work by Botvinick and colleagues). Some discussion of why DDMs were used and not MDPs would be helpful here, especially since the DDMs are not used to model planning per se but to more abstractly model an integration process.

- From what I can tell, choice data and reaction times were analyzed separately (the former with hierarchical probit models, the latter with DDM models). Would it be worthwhile to analyze them together in a DDM model? This type of analysis might be able to reveal speed/optimality tradeoffs.

**Have the authors made all data and (if applicable) computational code underlying the findings in their manuscript fully available?**

Reviewer #1: Yes

Reviewer #2: Yes

Reviewer #3: None

PLOS authors have the option to publish the peer review history of their article (what does this mean?). If published, this will include your full peer review and any attached files.

Reviewer #1: No

Reviewer #2: No

Reviewer #3: No
---

## [Decision Letter · Decision Letter 1]

29 Mar 2022

Dear Ms. Li,

Thank you very much for submitting your manuscript "A weighted constraint satisfaction approach to human goal-directed decision making" for consideration at PLOS Computational Biology. As with all papers reviewed by the journal, your manuscript was reviewed by members of the editorial board and by several independent reviewers. The reviewers appreciated the attention to an important topic. Based on the reviews, we are likely to accept this manuscript for publication, providing that you modify the manuscript according to the review recommendations.

Specifically, Reviewer 3 has still a concern about a perceptual confound. While I understand it is challenging to run a control experiment to rule out that confound, I would expect at least a discussion of that confound in the paper (and I agree with the reviewer that I also do not follow your logic for why it is not an issue). Once that is corrected, I think it is likely the paper can be accepted.

Sincerely,

Marieke Karlijn van Vugt, PhD

Associate Editor

PLOS Computational Biology

Natalia Komarova

Deputy Editor

PLOS Computational Biology

[LINK]

Reviewer's Responses to Questions

**Comments to the Authors:**

Reviewer #1: The revision is significantly improved and has addressed my concerns.

Reviewer #2: I thank the authors for fully engaging with my concerns as well as those of the other reviewers. The manuscript has been substantially rewritten and is much improved as a result. I have no outstanding or additional major comments and am happy to recommend the revised version for publication.

Reviewer #3: I appreciate the changes the authors have made to clarify aspects of the paper that were confusing. Below are my responses to the numbered points I made originally:

1. The authors agreed with the possible perceptual confound I pointed out, but they argue that this is unlikely to entirely explain their pattern of results. In support of this interpretation, they write, "Our finding that more accurate participants' path choices on the subgoal trials are less affected by the final goal does not rule out a perceptual salience effect, but does suggest that any such effect is subject to cognitive control, just as the Stroop effect and other task-dependent behaviors appear to be." I'm having a hard time following the logic of this sentence. For example, my understanding was that the effect of the final goal is being operationalized by measuring path choice accuracy/optimality, but they are indicating that this is a finding in itself (unless this refers to an analysis of RTs?). Is the idea that the pattern of responses for a subset of participants (the more accurate ones) indicates that attention to the goal is being modulated? If so, this needs to be spelled out more clearly in a response as well as in the paper itself.

I also want to re-emphasize my belief that although not strictly necessary, running control studies would be the cleanest and most direct way to rule out the perceptual confound. Such studies, if they show a weaker influence of a goal-like stimulus than an actual goal, would greatly strengthen and substantiate their central claim.

2. I appreciate the authors running this new analysis.

3. I have read the response to R1 that the authors mention and think it sounds sensible. It does however raise the question of what role the hierarchical decomposition theory is playing in this paper as it seems to be a straw model.

4. I appreciate that the authors have clarified this point.

**Have the authors made all data and (if applicable) computational code underlying the findings in their manuscript fully available?**

Reviewer #1: Yes

Reviewer #2: Yes

Reviewer #3: Yes

PLOS authors have the option to publish the peer review history of their article (what does this mean?). If published, this will include your full peer review and any attached files.

Reviewer #1: No

Reviewer #2: No

Reviewer #3: No

Figure Files:

Data Requirements:

Reproducibility:

References:

---

## [Editor Report · Decision Letter 2]

19 May 2022

Dear Ms. Li,

We are pleased to inform you that your manuscript 'A weighted constraint satisfaction approach to human goal-directed decision making' has been provisionally accepted for publication in PLOS Computational Biology.

Best regards,

Marieke Karlijn van Vugt, PhD

Associate Editor

PLOS Computational Biology

Natalia Komarova

Deputy Editor

PLOS Computational Biology

Thank you for submitting a revised version. Because two of the reviewers already accepted the manuscript, I myself have looked at the revision, and I think you satisfactorily address the concerns of the remaining reviewer on the interpretation of the data. Congratulations!

---

## [Editor Report · Acceptance letter]

9 Jun 2022

PCOMPBIOL-D-21-01858R2 

A weighted constraint satisfaction approach to human goal-directed decision making

Dear Dr Li,

I am pleased to inform you that your manuscript has been formally accepted for publication in PLOS Computational Biology. Your manuscript is now with our production department and you will be notified of the publication date in due course.

With kind regards,

Livia Horvath
